# Integrating the Ecological Security Pattern and the PLUS Model to Assess the Effects of Regional Ecological Restoration: A Case Study of Hefei City, Anhui Province

**DOI:** 10.3390/ijerph19116640

**Published:** 2022-05-29

**Authors:** Xiufeng Cao, Zhaoshun Liu, Shujie Li, Zhenjun Gao

**Affiliations:** College of Earth Sciences, Jilin University, Changchun 130061, China; caoxf20@mails.jlu.edu.cn (X.C.); lisj@jlu.edu.cn (S.L.); gaozj20@mails.jlu.edu.cn (Z.G.)

**Keywords:** ecological restoration, ecological security pattern, PLUS, MSPA, circuit theory, Hefei City

## Abstract

Most studies in the field of ecological restoration have only focused on repairing damaged land and have made no attempt to account for the impact of high-intensity land use on future landscape patterns. The purpose of this study was to propose a framework for evaluating the expected effects of ecological restoration based on land-use change and the ecological security pattern. Therefore, we integrated the PLUS model with the ecological security pattern and used Hefei City as a case study to conduct research. The results showed that from 2020 to 2030, land-use changes would occur primarily in the main urban area of Hefei and along the eastern shore of the Chaohu Lake watershed. Under the ecological protection scenario, arable land would be converted to construction land and woodland. Additionally, there would be an increase in ecological sources and pinch points in the area, and the number and area of the barriers would show a certain degree of reduction. The ecosystem quality, ecological integrity, and landscape connectivity of Hefei would be improved. This study offers a novel perspective for evaluating the expected effects of regional ecological restoration and provides an important reference for the dynamic formulation of multilevel ecological restoration policies.

## 1. Introduction

Rapid urbanization has become one of the defining characteristics of human social progress, and the country’s ecological environment is threatened by high-intensity land use and rapid land-use alterations. As a result, ensuring the structural and functional integrity of natural ecosystems for sustainable urban development is a global concern [1,2]. All countries must put forth effort into improving the structure and function of the global ecosystem [3]. Countries throughout the world have established technical schemes or planning methods to protect the ecological environment, including the urban growth boundary (UGB) [4], environmental network (EN) [5], and green infrastructure (GI) [6]. The ecological security pattern initiative, begun more recently in China, has steadily evolved into an essential planning and management tool for combining economic development with ecological protection and restoration [7]. However, few studies explore the correlation between land use and land cover (LULC), ecological security, and ecological restoration.

Land provides the material foundation for human survival and development. LULC elucidate the interplay between human society and the natural environment, which has emerged as a critical area of research on global environmental change. The research ranges from the implications of global climate change [8] to land change processes and their underlying mechanisms at various geographical scales, as well as ecological and environmental consequences. The study areas are predominantly concentrated in places with significant human–land conflicts, fragile biological conditions, and rapid population growth and urbanization [9]. Rapid urbanization is encroaching on land adjacent to dense human activity zones, such as arable land and woodland [10], resulting in a shift in land-use patterns. Continuous human activity rapidly accelerates the evolution of landscape structures, posing serious threats to landscape functions and ecosystems. This will disrupt the structure and function of the ecosystem [11], jeopardizing the landscape pattern and sustainability of the region [12]. Understanding and quantifying the spatiotemporal dynamics of LULC and their socio-ecological repercussions is critical for grasping the links between social and natural phenomena, particularly in urban regions. Consequently, assessing the evolution of land-use patterns is a prerequisite and foundation for analyzing local urban growth and ecological environmental conservation [13]. Simulating changes in land-use patterns scientifically and realistically through systematic land-use pattern simulation studies can help reveal the intrinsic connection between humans and the natural world [14].

To better comprehend, evaluate, and simulate land-use change, researchers have developed numerous LULC models, including the Cellular Automaton (CA) model, the CLUE-S model, the FLUS model, and the PLUS model. The CA model was the first and most popular model applied to predict future land-use spatial distribution [15]. Barredo et al. [16] integrated land-use factors with the CA model to simulate urban land use in Dublin City for the next 30 years. Xu et al. [17] coupled the random forest (RF) algorithm with the CA model to simulate the urban land allocation in Changzhou City in 2020. However, the traditional CA model is insufficient in terms of illuminating the underlying drivers of land-use change [18] and simulation accuracy [19]. Scholars have enhanced the traditional CA model from different perspectives. For instance, Verburg et al. [20] proposed a CLUE-S model based on system theory which was developed specifically for the regional-scale analysis of land use. The FLUS model introduced by Liu [21] improved the original CA model by incorporating self-adaptive inertia and a competition mechanism to process the complex competitions and interactions between the various land-use categories. Liang et al. [22] presented a patch-generating land-use simulation (PLUS) model. The PLUS model has been demonstrated to be a superior model that yields more accurate simulation results than the FLUS model, and it is more applicable for exploring the effects of climate change and human activities on future land-use dynamics [22,23].

The notion of ecological security pattern originates in landscape planning and is similar to the concepts of ecological networks and green infrastructure, all of which are aimed at protecting natural ecosystems within defined secure borders [24]. Regarded as an efficient tool to guarantee regional ecosystem security, the ecological security pattern is a spatial pattern that consists of certain critical locations and spatial linkages. Regional ecological processes can be effectively regulated by using the reciprocal feedback between ecological patterns and ecological processes. Effective regulatory instruments can ensure that ecological functions are carried out fully and that regional natural resources are allocated rationally, thereby contributing to the fulfilment of ecological security [25]. Establishing an ecological security pattern is one of the most essential strategies to alleviate the conflict between environmental conservation and economic development [24,26]. It is an inevitable choice to shift from ecological remediation after the destruction to ecological protection before the environment is destroyed. In the framework of a new era of ecological civilization, the formation of an ecological security pattern provides great support for China’s efforts to find ecological conservation measures that are more adapted to regional development demands. Territorial spatial planning under the concept of ecological civilization in China proposes a planning approach that emphasizes the priority of protecting the ecological environment called “reverse planning” or “anti-planning” [27]. Anti-planning is different from traditional planning, which places a high priority on economic growth and building grey infrastructure. Its planning procedure prioritizes the construction of an ecological security pattern and uses it to guide and restrict the development of the city. Anti-planning is a type of bottom-line thinking that prioritizes the protection of landscape components that provide important ecosystem functions.

The purpose of identifying crucial areas for ecological restoration based on the ecological security pattern is to maintain the ecosystem’s security and to ensure the long-term sustainable growth of the region. However, contemporary research on ecological restoration is focused on specialized viewpoints such as water ecology management [28], wetlands restoration [29], and erosion control [30]. These project-specific studies can produce positive outcomes for ecological restoration in specific areas, but the overall improvement in ecological function is still somewhat restricted. Additionally, current studies focus exclusively on restoring ecological space that has been destroyed [31] without considering the impact of future landscape pattern changes caused by high-intensity human activity. Our interventions rarely result in complete restoration, and uncertainty is to be expected in dealing with ecological restoration processes [32]. As a result, making dynamic adjustments in the implementation of the existing ecological restoration strategy is difficult. In this paper, we used the PLUS model to simulate land-use patterns in 2030 under the ecological restoration scenario with Hefei City as the study area. On this basis, we established ecological security patterns for 2020 and 2030 to identify crucial areas for ecological restoration, respectively. To figure out how well ecological restoration will work in this decade, we conducted a detailed comparative analysis of the crucial areas of restoration in 2020 and 2030. This would assist us in further defining the direction and scope of ecological restoration in the future.

The aim of this research was to present a framework for assessing the expected effects of ecological restoration. This framework incorporates a land-use simulation model and the ecological security pattern. It was designed as an analysis tool to further clarify the direction of the ecological restoration effort as well as the issues that require special attention. This study provided a new perspective for assessing the effectiveness of ecological restoration and for the formulation of multilevel ecological security policies. Dynamic assessment of future ecological concerns is more pertinent for regional sustainable development.

## 2. Study Area and Data Sources

### 2.1. Study Area

The study region for this article is Hefei City, Anhui Province (Figure 1). Hefei (116°41′–117°53′ E, 31°30′–32°28′ N), in central Anhui, is a natural hub of communications, being situated to the north of Chao Lake and standing on a low saddle crossing the northeastern extension of the Dabie Mountains, which form the divide between the Huai and Yangtze rivers. It has been the provincial capital since 1952, comprising four urban districts (Shushan, Luyang, Yaohai, and Baohe), one county-level city (Chaohu), and four counties (Changfeng, Feidong, Feixi, and Lujiang). Hefei has a northern subtropical monsoon climate with an average yearly temperature of 15.7 °C. Its precipitation averages 1000 mm/year, most of which occurs during May and August. The average altitude in Hefei is between 20 and 40 m above sea level. The total area of Hefei is 11,465 km^2^, with a municipal area of 1310 km^2^ and a main urban area of 2661 km^2^. According to the results of Hefei’s 7th National Census, the city had a resident population of 9,369,900 as of November 2020, putting it in the category of megacities.

### 2.2. Data Sources

The LULC data are derived from a dataset created by two Wuhan University professors. It contains data for 2000, 2010, and 2020, with a resolution of 30 m [33]. It is the most accurate land-use dataset currently available to the general public. Study area boundary data, annual average temperature data, soil type data, and GDP data are obtained from the Resource and Environment Science and Data Center (https://www.resdc.cn, accessed on 20 January 2022). Digital elevation model data were obtained from SRTMV4, with a resolution of 90 m (https://srtm.csi.cgiar.org, accessed on 1 February 2022). Population data were obtained from WorldPop; the resolution is 100 m (https://www.worldpop.org, accessed on 20 January 2022). The distance variables were extracted from the OpenStreetMap database (https://www.openstreetmap.org, accessed on 1 February 2022). Average annual precipitation data were obtained from the Institute of Mountain Hazards and Environment (http://www.imde.ac.cn/old, accessed on 5 February 2022), with a resolution of 30 m.

## 3. Methods

### 3.1. Simulation of Land Use

#### 3.1.1. The PLUS Model

The PLUS model is a patch-generating land-use simulation model that combines the advantages of both the transition analysis strategy (TAS) and the pattern analysis strategy (PAS). Additionally, multi-type random patch seeding is applied to model multiple land-use categories at fine resolution. The PLUS model can provide a better understanding of the relationships underlying land-use change. Compared with previous land-use models that are inadequate for determining the underlying drivers of land-use changes and cannot identify the temporal and spatial evolution of multiple land-use patches, the PLUS model can achieve better simulation accuracy and a more similar landscape layout [22].

As land-use change is complicated and non-linear, the selection of driving factors should be guided by the fundamental principles of comprehensiveness, data availability, and quantifiability. Besides natural forces, land-use change is also affected by economic and social variables, geography, and other things. With reference to existing research [23,34,35,36,37,38], we chose 15 factors that affect land-use expansion data to calculate the growth probability of each land-use type. These factors included 5 natural (elevation, aspect, temperature, precipitation, and slope), 3 socioeconomic (GDP density, night-time lighting, and population density), and 7 accessibility (distance to motorways, primary roads, secondary roads, tertiary roads, trunks, rails, and water). Neighborhood weights, as the important indicator in land-use simulation, represent the expansion intensity of land-use types. The neighborhood weight parameter ranges from 0 to 1, with a bigger value indicating a better expansion ability of the land type and a lower possibility of other land types occupying it. According to the actual land-use change of the study area from 2010 to 2020, the neighborhood weights were constantly adjusted. The simulation accuracy was assessed for different parameters, and the factor parameters with the best accuracy were picked. Therefore, the neighborhood weights were set as follows: arable land, 0.35; woodland, 0.16; grassland, 0.01; water, 0.05; wasteland, 0; construction land, 0.55.

The following process was used to simulate the land-use pattern in Hefei City: (1) We extracted regions that changed between 2010 and 2020 and then calculated the probability of development for each land type using a random forest algorithm; (2) we simulated the spatial pattern of future land use using the atlas of development probability; (3) we compared 2010 LULC data from the study area and development probabilities based on relevant parameters with simulation results for 2020 and real LULC data in the same year. Then we calculated the overall accuracy, Kappa coefficient, and figure of merit (FoM) to verify the accuracy of the model.

Hefei, as a representative of the new first-tier cities, still has much impetus and room for development in the short term. We first needed to determine the quantity of future land use in the future simulation. The Markov Chain model in this study is implemented to simulate the land-use demand from 2020 to 2030 based on the analysis of the land-use change during the 2010–2020 period. To highlight the critical significance of ecological restoration in ensuring regional ecological security, this study only considered the planning of territorial ecological restoration to project future scenario [39].

#### 3.1.2. Model Validation

To validate the model’s accuracy, we extracted land-use expansion data from 2000 to 2010, integrated it with the Markov Chain model to forecast land-use demand in 2020, simulated the land-use type map using the PLUS model, and compared it to the actual data. The overall accuracy, Kappa, and FoM coefficients were used to test the simulation accuracy of the PLUS model. We calculated the overall accuracy and Kappa coefficient by creating a confusion matrix of simulated and actual results for the raster cells. The values of Kappa are usually divided into 5 groups: 0~0.20 (slight), 0.21~0.40 (fair), 0.41~0.60 (moderate), 0.61~0.80 (substantial), and 0.81~1 (almost perfect) [40]. The Kappa coefficient is not very informative when the area undergoing change comprises a small proportion of the study area. Therefore, the FoM coefficient is constructed to assist in determining the accuracy of the simulation. FoM is superior to Kappa in assessing the accuracy of the simulation [41]. The calculation formula is as follows:(1)$$\mathrm{FoM}=\frac{B}{A+B+C+D}$$
where *A* is the error area where land use actually changes but is simulated as constant; *B* is the common area changing in both the actual map and simulations; *C* is the area that changed in both the actual and simulated maps, but the land-use types are different; *D* is the area that does not change in the actual map, while it changes during simulations.

### 3.2. Construstion of the Ecological Security Pattern

The ecological security pattern is a critical indicator for assessing the health and integrity of an ecosystem. The establishment of an ecological security pattern enables effective regional ecological preservation, a balance of ecological protection and economic development, and the maintenance of ecosystem services. It is an ecological network consisting of ecological sources, an ecological resistance surface, ecological corridors, and ecological crucial nodes. In this study, an ecological security pattern was constructed in the following steps: (1) identifying ecological sources; (2) constructing ecological resistance surface; (3) extracting ecological corridors; (4) identifying the ecological strategic nodes of ecological restoration based on circuit theory, including pinch points and barriers.

#### 3.2.1. Identifying Ecological Sources Based on MSPA

Ecological sources are generally stable habitat patches in the ecosystem which play an important role in promoting ecological processes, maintaining ecosystem integrity, and providing ecosystem services [42]. They serve as the foundation for the establishment of an ecological security pattern. To identify ecological sources, we employed morphological spatial pattern analysis (MSPA), an image processing method based on mathematical morphological principles [43]. It is capable of more precisely identifying the type and structure of a landscape. In this paper, land cover elements of high ecosystem service capacity, such as woodland, grassland, and water, were chosen as the foreground data for MSPA analysis. Arable land, construction land, and wasteland were used as the background data due to the lack of living environment for species to feed. Guidos Toolbox software was used to MSPA and then 7 non-overlapping landscape types were obtained. The core was extracted from the output result and then we used the Conefor software to evaluate landscape connectivity [44,45]. When it comes to determining landscape connectivity, the possible connectivity index (*PC*) and plaque importance index (*dPC_k_*) are frequently selected as crucial indicators since they can accurately indicate the degree of regional connectivity between core patches [46]. *PC* and *dPC_k_* were calculated as follows:(2)$${PC}_{}=\frac{\sum_{i=1}^{n}\sum_{j=1}^{n}a_{i}\times a_{j}\times p_{ij}^{*}}{A_{L}^{2}}=\frac{{PC}_{num}}{A_{L}^{2}}$$
(3)$${dPC}_{k}=100\times \frac{{PC}_{}-{PC}_{remove,k}}{PC}=100\times \frac{{\Delta PC}_{k}}{PC}$$
where *n* is the total amount of patches in the landscape; *a_i_* and *a_j_* are the attributes of patches *i* and *j*; $p_{ij}^{*}$ is the maximum product probability of all of the possible paths between patches *i* and *j*; *A_L_* corresponds to the total landscape area. *PC_remove,k_* is the metric value after the removal of *k*.

Considering patch radiation and landscape connectivity, we retained the patches with an area larger than 5 km^2^ and a *dPC* at or above 0.5 as ecological sources in this paper.

#### 3.2.2. Constructing Ecological Resistance Surfaces

The ecological resistance surface can be regarded as the degree of resistance of a land surface to the geographical dispersal and flow of species and ecological elements. Both natural conditions and anthropogenic disturbances have an effect on the ecological resistance surface. The types of landscape have a direct effect on the feasibility of those flows [47]. Topography can also have an impact on the dispersal of species, with the difficulty of dispersal increasing with elevation and slope [48]. We followed the principles of operability and data accessibility in selecting resistance factors. Considering the realities of the study, we set the resistance surface using representative indicators including land-use types, elevation, and slope [49].

#### 3.2.3. Extracting Ecological Corridors

Ecological corridors, as areas of low cumulative resistance between ecological source regions, allow species to migrate between habitats. Ecological corridors are typically composed of vegetation, water bodies, and other types of land cover element that provide ecosystem services [50]. Extraction of ecological corridors is essential for ecological flow and patch stability, as well as for the integrity of ecological functions and regional ecological security. Given the stochastic nature of ecological flow, circuit theory can accurately simulate ecological corridors between ecological sources, even in the absence of target species [51,52]. In this paper, we used the Linkage Mapper toolbox to extract corridors. We combined ecological sources and ecological resistance surfaces to build least-cost paths (LCPs) that connect several ecological sources as ecological corridors.

#### 3.2.4. Identifying Ecological Strategic Nodes

Ecological strategic nodes, colloquially referred to as “ecological stepping stones” are crucial for ensuring the sustainability of regional ecological functions. The identification and restoration of ecological strategic nodes can maintain the stability of ecological communities and enhance the regional ecosystem service. In this paper, we used the GIS tool Linkage Mapper to identify ecological strategic nodes based on circuit theory, including pinch points and barriers. The landscape is treated as a conductive surface in circuit theory, and random walks of electrons are used to simulate the ecological processes of species in the landscape. A finite value is assigned to each grid on a conductive surface that represents the difficulty of passing through that cell, and ecological sources are given the resistance value of 0.

Pinch points serve as crucial areas for species migration and energy transfers, which are critical for the connectivity of ecological security patterns [53]. We used the Pinchpoint Mapper to determine pinch points under the “all to one” mode. When 1 A current is injected from one node (an ecological source) and flows out from another, each passing grid is assigned a current value that represents the probability of the current passing through this grid. Pinch points are locations of high current density within an ecological network, indicating areas with a high probability of transit during migration. Since the corridor width would not affect the location and connectivity of pinch points, we set 45,000 m as the distance threshold [54].

Barriers are where species are impeded from migrating between ecological sources. According to circuit theory, the greater the resistance, the greater the impediment to species migration. Removing these areas with high resistance can significantly improve the connectivity between ecological sources [55]. We used Barrier Mapper, a plug-in included in the Linkage Mapper toolbox, to identify crucial barriers. The moving window approach was first employed to search for barriers with the following parameters: a minimum search radius of 100 m, a maximum search radius of 400 m, and a step size of 100 m [56,57]. Along with the ecological restoration of certain areas, the resistance of the places would be reduced. As a result, the cumulative resistance of the least-cost path connecting ecological sources through the recovered area could also be reduced. The restoring regions with the greatest cumulative resistance reduction were chosen as barriers.

## 4. Results

### 4.1. Simulation of Land-Use Pattern

The expansion potential of various types of land in Hefei was completely evaluated during the modelling process in this study. According to our hypothesis, the restoration measures were responsible for the changes in land use under the ecological restoration scenario. Therefore, we incorporated the requirements of territorial ecological restoration planning into the model as a constraint of anti-planning to simulate the land use in 2030. Based on the simulation results for 2030, we reconstructed the ecological security pattern and compared it with that of 2020. This paper provided a method to assess the expected effects of ecological restoration by coupling the PLUS model with the ecological security pattern. It can indicate the weaknesses and potential problems of the current ecological restoration plan, allowing the government to make dynamic policy modifications.

#### 4.1.1. Simulation Accuracy

The PLUS model was used in this study to forecast future land-use patterns in 2030 at a ten-year interval. It was necessary to simulate land use in 2020 using land-use changes between 2000 and 2010 and to evaluate the accuracy of the simulation against actual data. The Kappa coefficient, overall accuracy, and FoM value were calculated to determine the accuracy of simulation. The overall accuracy was 93.5%, with individual accuracy levels of more than 85% for arable land, water, and construction land. The Kappa coefficient of all land-use types was 0.85 (a value between 0.81 and 1, which means the result is almost perfect), and the FoM value was 0.45. When compared to the actual land-use pattern in 2020, the simulated results demonstrated good spatial consistency. The results revealed that the PLUS model was highly credible and that the model could be used to simulate the land-use pattern in 2030.

#### 4.1.2. Simulation Results

The classified LULC maps of Hefei were classified into six land-use types, namely arable land, grassland, wasteland, woodland, water, and construction land. Table 1 illustrates the total area of various land-use types in 2020 and 2030 as well as their coverage percentage. Arable land is the most common land-use type in Hefei. In terms of area, the area of arable land experiences a rapid decline from 2020 to 2030, where it decreased from 8725.04 km^2^ to 8170.89 km^2^ (73.485% to 71.269%). Correspondingly, the area of construction land increased by 213.95 km^2^ from 1362.71 km^2^ to 1576.66 km^2^, and the area of woodland increased by 38.34 km^2^ from 562.94 km^2^ to 601.29 km^2^. In contrast, the area of water remained largely steady. From the perspective of the spatial distribution (Figure 2), land-use changes are mainly distributed within the main urban areas of Hefei and in the eastern part of Chaohu Lake, which are the core development zones in Hefei’s Territorial Spatial Planning. Between 2020 and 2030, the various processes of land-use change would be primarily concentrated near the built-up land border. When the simulation results in 2030 are compared with the actual land use in 2020, it is clear that the area of woodland in the main urban areas of Hefei has obviously increased. In general, the ecological space in Hefei would be expanded under the ecological restoration scenario.

### 4.2. Ecological Security Pattern

#### 4.2.1. Ecological Sources Analysis

Under the criterion of patches with an area larger than 5 km^2^ and a dPC at or above 0.5, 19 and 16 ecological sources were identified in 2020 and 2030 in Hefei, and the corresponding land-use types are shown in Table 2. In 2020, the area of identified ecological sources was 1077.65 km^2^, accounting for 9.40% of the total land area of Hefei. In 2030, the area of identified ecological sources was 1099.59 km^2^, accounting for 9.59% of the total land area of Hefei. The land-use types of ecological sources in Hefei consisted of woodland, grassland, and water, all of which had ecosystem services, with the water area accounting for more than 70% of the total area of ecological sources. Chaohu Lake was the largest ecological source, covering over half of the total area. In the period from 2020 to 2030, two separate ecological sources in Feixi County were expanded and merged into one. From the perspective of the spatial distribution (Figure 3), ecological sources were distributed in blocks in the outer suburbs of the study area, particularly in the south and central-eastern regions. The spatial distribution of ecological sources in Hefei was extremely imbalanced, mainly concentrated in Chaohu City, while those in the remaining counties were small and scattered. Due to urban expansion, ecological sources in the central area of Lujiang County and on the eastern shore of Chaohu Lake would be encroached upon and removed from ecological sources by 2030, as they cannot meet the screening criteria (shown in red frame).

#### 4.2.2. Integrated Ecological Resistance Surface

Both natural conditions and anthropogenic disturbance influence the ecological processes of species migration and energy flow. Using the method described in Section 3.2.2, the comprehensive ecological resistance surfaces for Hefei in 2020 and 2030 were established based on land-use data from the respective years, slope, and topographic relief (Table 3). As illustrated in Figure 4, high ecological resistance values are concentrated within the main urban areas of Hefei and the eastern portion of the Chaohu Lake watershed, forming a pattern of “centrally dense and distributed around”, and the resistance surface’s area of high values has a propensity to expand. “Centrally dense” mainly represented the main urban areas, which had high-density built-up and a large population and were susceptible to anthropogenic disturbances. The ecological resistance of arable land is relatively higher. Despite the fact that arable land has a high level of vegetation cover, single species patterns in vegetation will inhibit the growth of other species. Watersheds and woodland were identified as regions with low ecological resistance values, and there were few buffer zones between high and low resistance ones. When compared to the spatial distribution of ecological sources, the lack of effective links between areas of high ecological resistance and ecological sources can be demonstrated, which would impede the migration of species and the flow of energy. In summary, the expansion of construction land in Hefei will lead to a parallel increase in the area of zones with high ecological resistance values, which would be detrimental to the sustainable development of the regions.

#### 4.2.3. Spatial Pattern of Ecological Corridors

In this study, the ecological sources and comprehensive ecological resistance surface were used to extract ecological corridors (Figure 5), including primary and secondary ones. In 2020, 29 primary corridors with a total length of 605.38 km and 15 secondary corridors with a total length of 751.90 km were identified. In 2030, 21 primary corridors with a cumulative length of 471.43 km and 16 secondary corridors with a total length of 824.98 km were identified. The distribution of ecological corridors generally avoided built-up land and followed areas with low ecological resistance, such as rivers and woodland. The cumulative length of the corridors decreased between 2020 and 2030. In 2030, the location of primary corridor No. 3 and secondary corridor No. 6 shifted, and the primary corridors No. 4 and No. 5 disappeared. The reason for this phenomenon was that the expansion of built-up land has weakened the ecological carrying capacity of these areas. The primary corridors No. 1 and No. 2 would not change significantly, as they are the main rivers in Hefei. Ecological corridors in Hefei could be characterized by spatial heterogeneity. To be specific, ecological corridors are more densely dispersed in the study area’s central and eastern regions. These regions were abundant in natural resources with better ecosystem quality, such as water and woodland, which provided the ideal conditions for species migration.

#### 4.2.4. Ecological Strategic Nodes

In this paper, the application of Jenks Natural Breaks helped to identify ecological strategic nodes, including pinch points and barriers. As shown in Figure 6, the maximum cumulative current values are 0.862 and 0.971 in 2020 and 2030, respectively. By assigning a grade to the cumulative current, 31 pinch points totaling 10.51 km^2^ in 2020 and 45 pinch points totaling 18.13 km^2^ in 2030 were identified in the study area. In terms of land-use types, the pinch points both in 2020 and 2030 were dominated by water and woodland, which accounted for over 80% of the total. The remaining land-use types that made up the pinch points were arable land and a small amount of construction land. Most of the pinch points identified were natural water corridors, so they exhibited a narrow strip-like distribution. The migration of species was confined to such small areas for the reason that the natural aquatic corridors were surrounded by built-up and arable land that was heavily influenced by anthropogenic activities. From 2020 to 2030, the number and area of pinch points in the study area would increase, mainly distributed on the east shore of Chaohu Lake, the central portions of Lujiang County, and the southeastern portions of Changfeng County. The results showed that ecological connectivity would be significantly improved and habitat quality would be enhanced.

As depicted in Figure 7, the maximum cumulative current recovery values are 419.416 and 404.894 in 2020 and 2030, respectively. The distribution characteristics displayed that the areas with high cumulative current recovery values were clustered within the main urban areas. Classifying the cumulative current recovery values using Jenks Natural Breaks, 35 barriers with a total area of 59.34 km^2^ in 2020 and 21 barriers with a total area of 33.12 km^2^ in 2030 were identified in the study area. The number and area of barriers would be considerably reduced over the next decade, especially in the main urban areas of the study area. The identified barriers were primarily comprised of arable land and construction land, which together accounted for more than 90% of the total area of the recognized ones, and the barriers exhibited a more regular morphological character than pinch points. The dispersion of barriers in 2030 tended to be more concentrated. In 2030, there were 20 ecological barriers located in Hefei’s main urban areas and one outside. However, there were nine barriers beyond the main urban zones of Hefei in 2020. A comparison of land-use data between 2020 and 2030 revealed that the barriers identified in 2020 would be well recovered, with considerable quantities of woodland and water occurring within these areas.

## 5. Discussion

### 5.1. Comparison of Related Studies

The expansion of urban areas destroys ecological structures and components, posing a severe threat to ecosystems. There has been research suggesting that changing urban patterns and forms can have different effects on the environment [58]. Several studies have been undertaken to mitigate the detrimental effects of urban expansion on ecosystems by adopting the spatial distribution of ecological components and functions [59]. Selecting relevant indicators of ecological constraint is a precondition for performing land use simulation under ecological constraints. In accordance with the development planning and ecological protection requirements of Hefei, this study employed natural, socioeconomic, and accessibility as the constraining factors of land-use change to model the expansion possibilities of land-use type. Then, using the CARS module, simulate the land-use pattern in 2030 based on the ecological restoration scenario. Previous research has demonstrated that the PLUS model can simulate multiple types of land-use change at the patch level with more precision and a more realistic landscape pattern [22,60]. Thus, using the PLUS model for land-use simulation in this study is highly suitable.

An analysis of the changes in land-use types in Hefei revealed a significant reduction in the area of arable land, the majority of which had been converted to construction land and woodland. Consistent with the findings of Janeczko et al., there is a greater chance that the arable land surrounding the city will be converted into construction land [61]. Land use changes were mainly concentrated in the main urban area of Hefei and on the eastern shore of the Chaohu Lake watershed, which has greater development prospects in the future. Under ecological restrictions, additional arable land would be absorbed by urban expansion as an alternative to ecological land. The regularity of land-use patterns under ecological limitations corroborated earlier research that found ecological constraints to be beneficial in improving urban land compactness [62,63]. The most plausible explanation for this situation is the abundance of arable land that surrounds urban areas. Although ecological restrictions can help offset ecological degradation to a degree, they come at the cost of less arable land [64]. Consequently, measures limiting the extent of urban land should be promoted as well, for example, by establishing urban growth boundaries to constrain urban expansion or by conducting land consolidation projects to capitalize on the inherent development potential of cities.

The term “ecological security pattern” refers to a possible spatial pattern of ecological systems within the landscape [25]. It is feasible to successfully regulate ecological processes and eventually attain regional ecological security by establishing an ecological security pattern. Existing research has concentrated on the establishment of ecological corridors [65], the delineation of ecological redlines [66], and the identification of ecological threats [67]. However, the majority of studies are focused on the current pattern of geographical features, omitting changes in land use caused by natural and human forces. In recent years, some scholars have begun to focus on the application and optimization of ecological security patterns from the perspective of land use changes. Zhang et al. [68] and Li et al. [69] simulated the land use pattern of the study area in 2025 under different scenarios and constructed the ecological security pattern. We should think more about whether or not the ecological security pattern as currently established will be relevant in the future. Consequently, we coupled the PLUS model with the ecological security pattern to analyze the expected effects of regional ecological restoration. We take Hefei, a representative of the new first-tier cities, as an example to conduct our study, which is applicable to the actual situation of urban development in China.

The main goal of the current study was to establish and thoroughly analyze the ecological security patterns for 2020 and 2030. Hefei would continue to face significant urban development pressure over the following decade, while the area of ecological sources would expand somewhat, from 1077.65 km^2^ to 1099.59 km^2^. Ecological corridors in Hefei were mainly composed of water bodies and woodlands with ecosystem services, providing ideal conditions for species migration, and they exhibited great spatial heterogeneity. The area and number of pinch points would increase, and they would be distributed in a narrow stripe-like pattern. The pinch points would be extremely vulnerable to the threat of anthropogenic activities due to their distinctive spatial distribution. Additionally, there would be a significant reduction in barriers, and their distribution would tend to be more concentrated in the main urban areas. They were the most prominent impediments to landscape connectivity, and removing them would help to improve ecological connectivity. However, it would be difficult to implement ecological restoration projects in these areas that are the economic support of Hefei. By identifying the expected effects of ecological restoration in the Hefei area and actively addressing potential challenges, the interference of human activities in the ecological restoration process could be reduced. Identifying potential ecological hazards in advance and consciously enhancing ecological protection would be meaningful to improving the regional ecosystem quality. The establishment of a regional ecological security pattern can substantially alleviate tension caused by fast urbanization, thereby ensuring regional sustainability.

### 5.2. Proposals for Further Strengthening Ecological Restoration in the Next Decade

In this study, the ecological security pattern of Hefei in 2030 was constructed and compared with that of 2020 using the simulated land-use pattern under the ecological protection scenario. It could be found that ecological restoration effectively improved ecosystem quality, but some aspects needed to be enhanced. Based on the current assessment, we made some recommendations.

Pinch points should be treated as ecological priority protection zones. They are recommended for conservation employing natural restoration, supplemented by human restoration. Harnessing the power of nature itself to improve ecosystem services and ecological security. The conservation of pinch points should focus on maintaining the stability of the quantity, structure, and function of the ecological space. It is essential to provide a good ecological space for the migration, habitat, and reproduction of species. Given that the pinch points are mainly woodland and water, more resilient land policies should be implemented to prevent the degradation of watersheds and woodlands.

The barriers are primarily composed of construction and arable land, and they typically exhibit a high level of ecological resistance. Elimination or reduction of existing barriers has the potential to significantly enhance landscape connectivity and ecological function. Consequently, they should be treated as the key area for ecological restoration and can be gradually improved through nature-based solutions [70]. The limitation of ecological barriers should be linked to spatial policy and planning in the urban landscape in an attempt to achieve a land-use configuration that will balance urban development needs and environmental requirements. Concerning ecological patches within urban construction, the ecological functions of urban green space, forest parks, and other areas should be utilized to their full potential to improve the regional ecological environment. To promote ecological connectivity, it is essential to build protective forest belts along the margins of arable land. The process of barriers’ reduction should not be restricted to the immediate needs of the situation, but should also address the elimination of ecological damage that future urban development may cause.

### 5.3. Uncertainties and Prospects

In this study, we present a framework for assessing the expected effects of ecological restoration, which is verified using the case study of Hefei. This study is exploratory in nature, but it also has some limitations. Land use patterns are influenced by different natural and socioeconomic factors at varying temporal and spatial scales. Considering the data availability, we only take some driving factors into account that are closely relevant to the purpose of our study to simulate land use patterns. Therefore, future research should be undertaken to explore how to refine the indicator hierarchy. The most difficult aspect of modeling a resistance surface is determining resistance factors and assigning resistance values to distinct indicators because the actual effects of diversity gradients on migration, survival, and reproduction are generally unpredictable [71]. Due to the simulation, it is difficult to support the selection of specific factors like fractional vegetation cover. Existing studies do not have uniform selection criteria, so we use land-use type, slope, and topographic relief as resistance indicators to construct ecological resistance surfaces.

## 6. Conclusions

The spatial ecological security pattern of the territory depicts an optimized ecological network, which can scientifically define the spatial relationship between high-intensity human activities and ecological restoration. The goal of restoration is to establish self-sustaining and resilient systems; therefore, they must be compatible with their surrounding environmental context and landscape pattern. To compensate for the lack of consideration of future landscape change in existing ecological restoration studies. Therefore, in this study, the PLUS model, as a more suitable model for simulating the dynamics of human activities on land use, has been innovatively coupled with the ecological security pattern. Hefei, a representative of the new first-tier cities with strong growth momentum, was chosen as the subject of our work, which is consistent with the actual urban development in China. Our research could provide a reference for balancing urban development and ecological restoration in other new first-tier cities, including Hangzhou, Changsha, and Wuhan.

In this study, the ecological strategic nodes that need to be restored in Hefei were first identified based on the land use pattern in 2020. According to the development and environmental protection requirements, the restoration of identified areas was then incorporated into the PLUS model as the planning constraint condition to simulate the land-use pattern of Hefei in 2030. Finally, the ecological strategic nodes were identified based on the simulation results in 2030, and they were compared with those in 2020. We further clarified the direction of the ecological restoration effort, as well as the issues that required special attention. The results showed that:(1)From 2020 to 2030, land-use changes would occur primarily in the main urban area of Hefei and along the eastern shore of the Chaohu Lake watershed. Under the ecological protection scenario, large amounts of arable land would be converted to construction land and woodland.(2)There was an increase in the area of ecological sources and pinch points from 2020 to 2030, and a notable reduction in the number and area of barriers. Overall, these results indicated that the ecosystem quality, ecological integrity, and landscape connectivity of Hefei would be considerably improved.

The study certainly adds to our understanding of the relationship between future landscape patterns and ecological restoration. Aside from its exploratory nature, this paper offers some insight into assessing the intended effect of ecological restoration and provides a new perspective for the formulation of multilevel ecological restoration policies. There is no doubt that land use change-based modeling provides a good direction for the dynamic adjustment of ecological security patterns.

## Figures and Tables

**Figure 1 ijerph-19-06640-f001:**
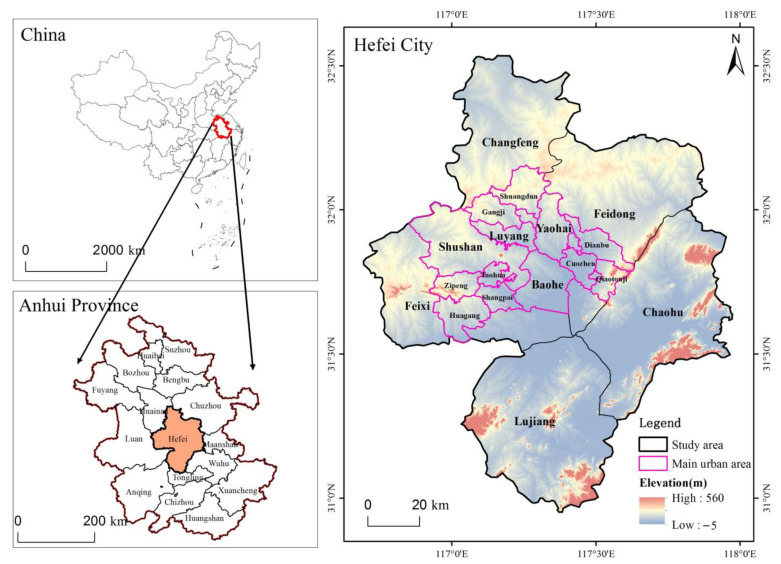
The spatial location of the study area.

**Figure 2 ijerph-19-06640-f002:**
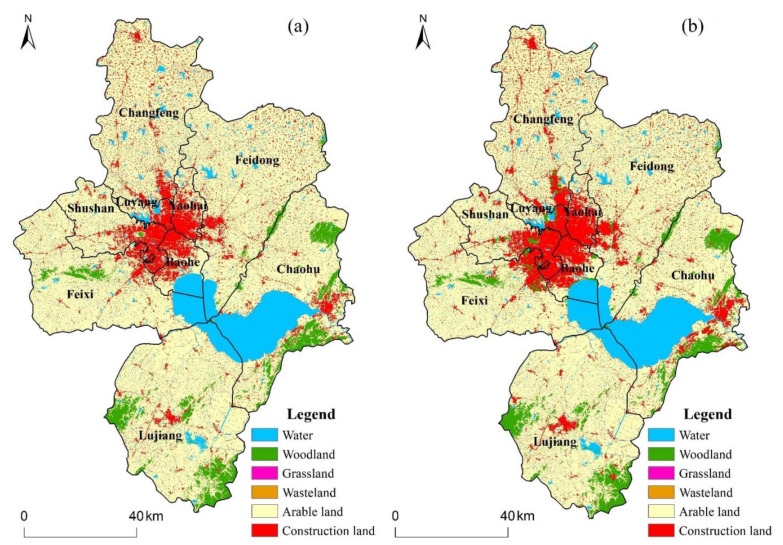
LULC classification maps of Hefei for (**a**) 2020, (**b**) 2030.

**Figure 3 ijerph-19-06640-f003:**
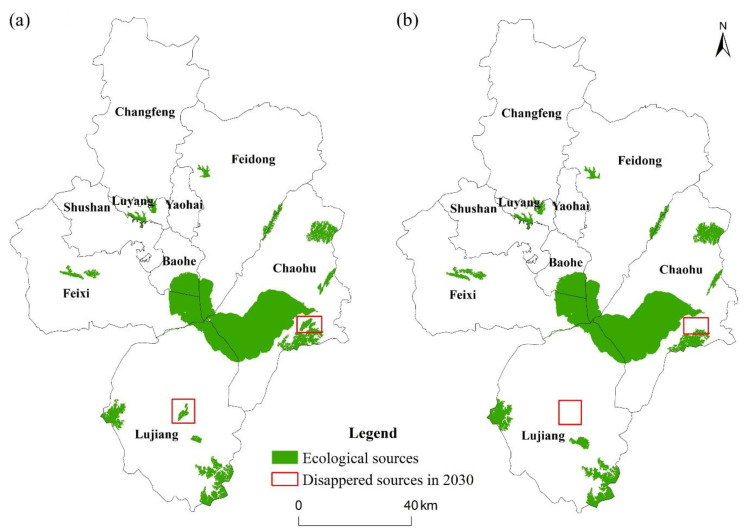
Ecological sources of Hefei for (**a**) 2020 and (**b**) 2030. Note: The areas depicted by the red frames are ecological sources in 2020, and the portions that would be removed in 2030 because they failed to meet the ecological source screening criteria.

**Figure 4 ijerph-19-06640-f004:**
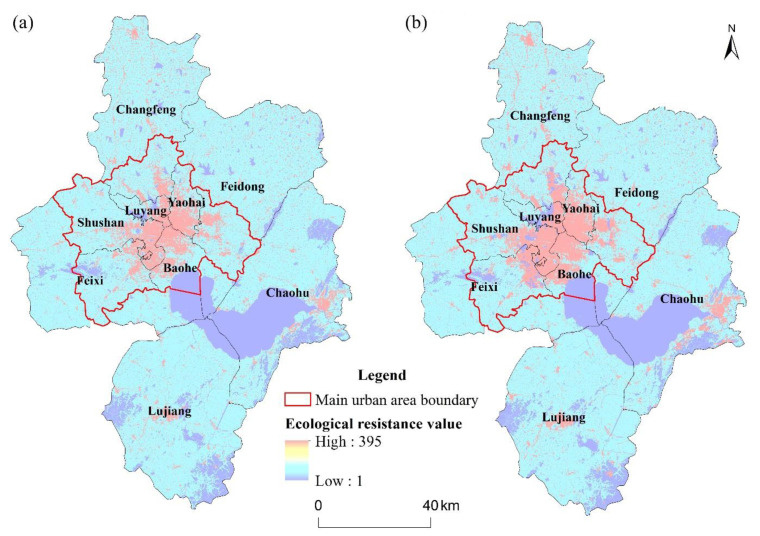
Ecological resistance surface of Hefei for (**a**) 2020 and (**b**) 2030.

**Figure 5 ijerph-19-06640-f005:**
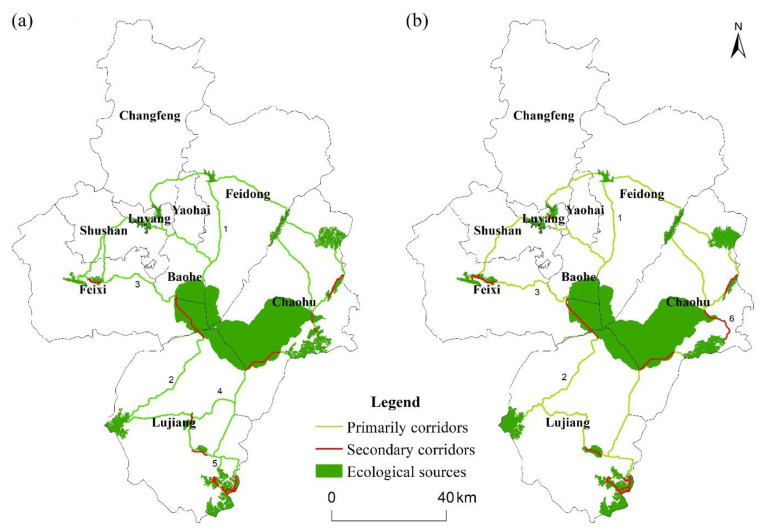
Ecological corridors of Hefei for (**a**) 2020, (**b**) 2030.

**Figure 6 ijerph-19-06640-f006:**
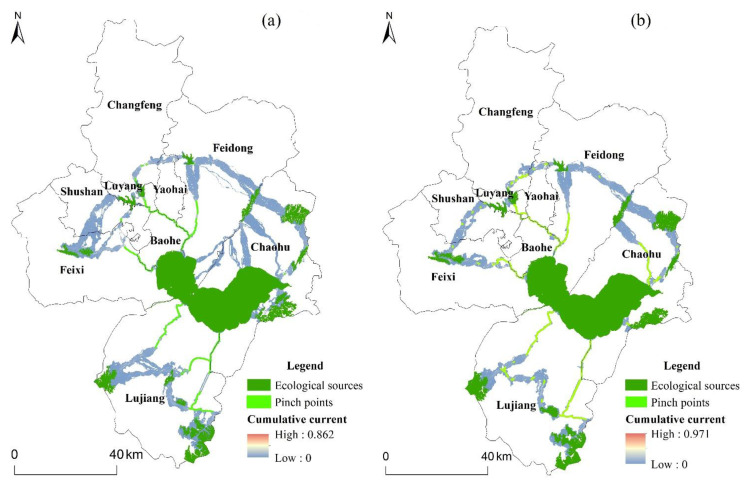
Pinch points of Hefei for (**a**) 2020 and (**b**) 2030.

**Figure 7 ijerph-19-06640-f007:**
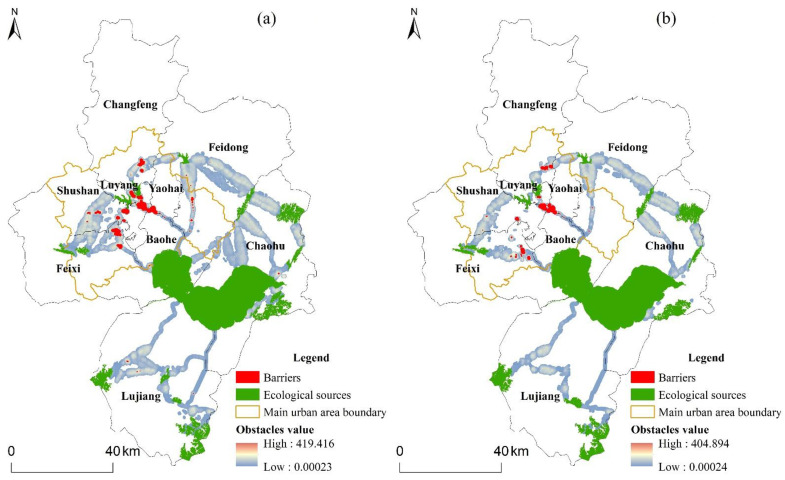
Barriers of Hefei for (**a**) 2020 and (**b**) 2030.

**Table 1 ijerph-19-06640-t001:** Land-use area and percentage in Hefei.

Area and Proportion	Land-Use Type (km^2^/%)					
	Arable Land	Woodland	Grassland	Water	Wasteland	Construction Land
2020	8425.04	562.94	0.48	1113.69	0.06	1362.71
	(73.485)	(4.910)	(0.004)	(9.714)	(0.001)	(11.886)
2030	8170.89	601.29	0.38	1115.67	0.03	1576.66
	(71.269)	(5.245)	(0.003)	(9.731)	(0.001)	(13.752)

Note: The percentages in parentheses are the proportions of total land area for each land-use type.

**Table 2 ijerph-19-06640-t002:** Land-use area and percentage of ecological sources in Hefei.

Area and Proportion	Land-Use Type (km^2^/%)			
	Woodland	Grassland	Water	Total
2020	270.41 (25.09)	0.0045	807.23 (74.91)	1077.65
2030	279.47 (25.42)	0.0009	820.12 (74.58)	1099.59

Note: The percentage in parentheses represents the proportion of the total area of ecological sources occupied by each land-use category.

**Table 3 ijerph-19-06640-t003:** Resistance value for each resistance factor.

Resistance Indicator	Weight	Resistance Factor	Resistance Coefficient
Land-use type	0.7	Woodland	1
		Water	10
		Arable land	100
		Grassland	100
		Wasteland	300
		Construction land	500
Slope (°)	0.2	[0, 8)	1
		[8, 15)	10
		[15, 25)	50
		[25, 35)	100
		[35, 53)	200
Topographic relief (m)	0.1	[0, 25)	1
		[25, 50)	10
		[50, 100)	50
		[100, 200)	100
		[200, 471)	200

## Data Availability

Not applicable.

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
