# Peer review of "Integrating the Ecological Security Pattern and the PLUS Model to Assess the Effects of Regional Ecological Restoration: A Case Study of Hefei City, Anhui Province"

_ijerph, 2022, doi:10.3390/ijerph19116640_

Round 1
Reviewer 1 Report
Paper is very well prepared and completely understandable. Abstract gives insight into all relevant parts of the research. There is some minor comments which should be addressed;
Lines 130-133 - add more references beside 17 and 18
Please explain more weight setting for the neighborhood (lines 148-150)
Authors should refer to the importance of the research on the global context.
Reviewer 2 Report
Manuscript ID: ijerph-1725214
Type of manuscript: Article
Title: Integrating the Ecological Security Pattern and the PLUS Model to Assess the Effects of Regional Ecological Restoration: A Case Study of Hefei City, Anhui Province
Authors: Xiufeng Cao, Zhaoshun Liu*, Shujie Li, Zhenjun Gao
Comments and Suggestions for Authors:
The topic and the idea of the study are valuable and the main assumptions are valuable, also corresponds very well to the contemporary situation of areas LULC and their changes. However some elements of the manuscript should be improved.
1. The title is quite long but clear, and in line with the presented study.
2. Abstract is generally well constructed and presents main elements of the study. However, I would avoid the direct information e.g. that there were notable reduction in the number and area of barriers. The results presented in the study show rather the possible tendencies, so it should be called in a bit more conservative wording, etc.
3. Key words are related to the topic.
4. The background of the study presented in the section of Introduction is important, and focuses on the most important aspects and information related to the study. Some more literature items could be cited.
5. The aim of the study is not presented (?).In the last paragraph of the Introduction section, the Authors generally write about the research itself, especially about the used the PLUS model and a bit about its argumentation, but not about its purpose. The main objective of the study must be clearly formulated, clearly stated and defined, with a short justification - this needs to be completed / corrected.
6. The study area as well as data sources are shortly and clearly presented.
7. The presentation and explanation of methods, including 3.1. Simulation of Land Use with the Model Validation, also the 3.2. Construction of the Ecological Security Pattern, are quite well presented, also division into subsections is understandable.
8. The Results – the order of results presentation is generally well organized and divided into parts which follow the stages of the study. Figures follow presented data and description.
However, there are some mistakes – e.g. in Table 2. “Land use area and percentage of ecological sources in Hefei” – the percentage of Water in 2030 is incorrect – it is higher than in 2020 regarding the increasing area covered by water (from 807.23 to 820.12). Thus, this mistake is lowering the quality of the study and this section must be revised.
Also part 4.2.2. Integrated Ecological Resistance Surface must be revised – the division of percentage presented in Table 3 is insufficient - the Resistance Factors can not include the same amount in 2 different indicators, e.g. if one is 0°~8°, the next one should be from 9°~15° , etc. – it applies to both: Slope, and Topographic relief. This approach is unscientific and imprecise, thus distorting the obtained results and, consequently, undermining the quality of the study. This important part of results presentation must be revised/improved.
9. The section of Discussion is quite developed, mainly in the context of comparison with other studies used similar methods, however, some more comparative studies should be introduced to extend the scientific soundness of this approach, to argue more the value of used methods in the process of forecasting changes of the area.
As the purpose of the study (simulation) was not clearly defined at the beginning of the manuscript, its reference is unclear. The part of Discussion related to the purpose needs to be specified, also very precise and directly connected with the objective of the study.
The approach /formulating of words/ in lines 471 to 481 is not clear – “restoration” of barriers such as increasing number of built areas generally can’t be found as a positive approach in the context of the development of green/natural corridors; the presented context should rather means to connect the development of built areas with implementation of more natural elements, also continuous forms and linked to natural areas - to plan them (if they can not be introduced as natural processes), etc.
Discussion should generally highlight more the value of individual study done by Authors and its role for the development of the studied region.
10. In the section of Conclusions Authors should also explain more and clearly why the simulation presented in the study is important/its novelty; maybe also what is not only theoretical but also practical dimension of the obtained results for the studied area.
Others:
- the English language needs to be improved, some sentences require a proofreading to increase the quality of the presentation of studied contexts and/or the research itself, also some unclear wording is used, e.g. it is impossible to “install” green spaces, etc. – it sounds unprofessional.
- Figure 2 – it is needed to add to the Figure description the explanation what means the red frame used on the picture
- there are some missed spaces in the text, e.g. lines 113, 295, 318, in top of Figure 4, 341, 381, Figure 6 and 7, 399, etc.
- some typos in the text
Reviewer 3 Report
Dear Editor
As soon as I received the article, I carefully studied it.
I would like to thank the authors for this article they wrote with their efforts.
I would like to state that it is a good work and that I have obtained useful information.
I think that the reorganization of the article, taking into account the criticisms
I have mentioned below, will further increase its quality.
I evaluated that the summary writing method discussed by the authors in the abstract part of the article would not reflect a scientific research article well. For this reason, I think that the abstract should be reconsidered and discussed in a way that presents the definition of the problem in the article, the purpose of the study, the most important result obtained by the authors, and an important suggestion.
Reviewer 4 Report
Thank you for the opportunity to review this text. The search for a pattern of urban environmental safety is an issue of great importance to the health of urban residents. It can be assumed with a high degree of probability that the arable land surrounding cities will eventually be converted into building land and, to a lesser extent, forest land. Therefore, it is necessary to determine the limits to which urbanization will go. I recommend the attention of the authors of the article DOI:10.3390/su11113007, in which the authors also reached a similar conclusion when analyzing the development of the city in recent decades. The article is very well structured. The methodology is well described and the results are graphically clear. My basically only comment is on the terminology used for landscape. Well, land use structure is not exactly landscape. Therefore I suggest replacing line 185 in section 3.2.1 and moving away from the term landscape to land cover elements. And this comment basically applies to the rest of the landscape emphasis in this paper.
Round 2
Reviewer 2 Report
Manuscript ID: ijerph-1725214
Title: Integrating the Ecological Security Pattern and the PLUS Model to Assess the Effects of Regional Ecological Restoration: A Case Study of Hefei City, Anhui Province
I appreciate the work of Authors and changes introduced to the manuscript which increased its scientific soundness much.
Last important suggestion to be implemented:
Authors still use the word ‘restoration’ connected to barriers while correct are other words: ‘elimination’, ‘reduction’ or ‘limitation’ of barriers in the context of obtaining new spaces with ecological functions. The idea of the research is related to ecological restoration but not restoration of barriers! This mistake of meaning is existing only in one paragraph (in lines: 537-550), thus Authors should check the translation in the part of Discussion.
Summing up, the meaning of this paragraph is different at present form than expected as a result of the study. It should sounds such as:
“The barriers are primarily composed of construction and arable land, and they typically exhibit a high level of ecological resistance. Restoring Elimination or reduction of existing barriers has the potential to significantly improve landscape connectivity and ecological function. Consequently, they should be treated as the key area for ecological restoration and can be gradually improved by nature-based solutions [71]. The restoration limitation of ecological barriers should be linked to spatial policy and planning in the urban landscape in an attempt to achieve a land use configuration that will balance urban development needs and environmental requirements. Concerning ecological patches within urban construction, the ecological functions of urban green space, forest parks, and other areas should be utilized to their full potential to improve the regional ecological environment. To improve ecological connectivity, it is essential to build protective forest belts along the margins of arable land. Restoration The process of barriers' reduction should not be limited to the immediate needs of the situation, but should also address the elimination of ecological damage that future urban development may cause.”
Others:
Some words are repeated many times in following sentences.
